# Grape and Wine Metabolomics to Develop New Insights Using Untargeted and Targeted Approaches

**Farhana R Pinu** 

The New Zealand Institute for Plant and Food Research Limited, Private Bag 92169, Auckland 1142, New Zealand; farhana.pinu@plantandfood.co.nz; Tel.: +64-9926-3565

**Abstract:** Chemical analysis of grape juice and wine has been performed for over 50 years in a targeted manner to determine a limited number of compounds using Gas Chromatography, Mass-Spectrometry (GC-MS) and High Pressure Liquid Chromatography (HPLC). Therefore, it only allowed the determination of metabolites that are present in high concentration, including major sugars, amino acids and some important carboxylic acids. Thus, the roles of many significant but less concentrated metabolites during wine making process are still not known. This is where metabolomics shows its enormous potential, mainly because of its capability in analyzing over 1000 metabolites in a single run due to the recent advancements of high resolution and sensitive analytical instruments. Metabolomics has predominantly been adopted by many wine scientists as a hypothesis-generating tool in an unbiased and non-targeted way to address various issues, including characterization of geographical origin (*terroir*) and wine yeast metabolic traits, determination of biomarkers for aroma compounds, and the monitoring of growth developments of grape vines and grapes. The aim of this review is to explore the published literature that made use of both targeted and untargeted metabolomics to study grapes and wines and also the fermentation process. In addition, insights are also provided into many other possible avenues where metabolomics shows tremendous potential as a question-driven approach in grape and wine research.

**Keywords:** winemaking; metabolite profiling; non-targeted analysis; classical chemical analysis; metabolic modelling; yeast physiology and metabolism; vineyard management

## 1. Introduction

Targeted metabolite analysis of grape juice and wine has been carried out for a long time, specifically after the development of gas chromatography and mass spectrometry (GC-MS) [1]. Most of these studies were performed to determine the variety of wine based on its aroma composition [2–8]. Some studies have also focused on the overall composition of grape juice and determined mainly the amount of sugars, amino acids and some important carboxylic acids using different enzymatic methods or high pressure liquid chromatography (HPLC) [9–12]. By using classical chemical analytical methods, it was only possible to determine the specific groups of metabolites that were present usually in high concentration in both grape juices and wines. The unavailability of appropriate analytical instruments and suitable methods to determine the concentrations of lower abundant metabolites were the main reasons behind this scenario [13] Therefore, the exact contribution of many significant but low-concentration metabolites to the wine fermentation process was not identified. On the contrary, comprehensive and unbiased approaches of metabolomics are now providing thorough information about many different groups of compounds in grape juices and wines and are therefore more advantageous than traditional targeted analysis [13,14].

As one of the most newly introduced "-omic" technologies, metabolomics was initially proposed as a tool in functional genomics [14]. The other "omics", technologies: genomics, transcriptomics and

proteomics are focused on genes, RNA and proteins, respectively, whereas metabolomics is the study of the most downstream products of cells called metabolites [13] (Figure 1). Metabolomics is typically known as an unbiased, non-targeted and holistic analysis of cell metabolites [15]. However, application of targeted metabolomics analysis is also on the rise [16–19]. It is now an emerging research area with application in different fields including functional genomics and systems biology [14,18,20–26].

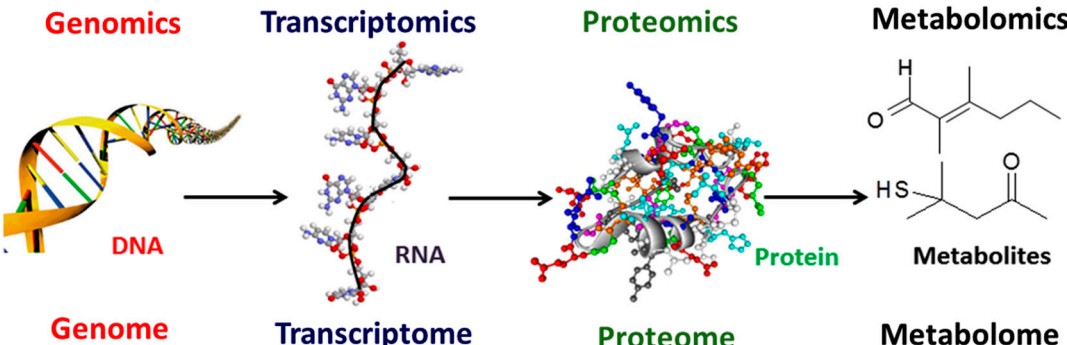

**Figure 1.** The hierarchy of "omics" technologies. The post-genomics approaches (e.g., transcriptomics, proteomics, and metabolomics) together can provide comprehensive information and better understanding about the biological system (adopted from Pinu [13]).

As a question and data-driven approach, metabolomics already shows tremendous potential in food and agricultural sciences although the application of metabolomics has started just over a decade ago [27,28]. Within this time period, it has proven to be an important and powerful approach and has been used to analyze metabolites in agricultural (and food) products in both targeted and untargeted ways [16,29–33]. Like other food and agricultural products, the introduction of metabolomics in grape and wine research also garnered considerable attraction, mostly as a hypothesis-generating tool [30,34–39]. The main aim of this review is to re-visit the available published literature where either targeted or untargeted metabolomics has been applied to study grapes, wines and microorganisms associated with winemaking. Existing challenges and ways to overcome those are also provided in addition to discussing the future perspectives of metabolomics in grape and wine research.

## 2. Advancements in Metabolomics as an Emerging Tool within the Last Decade

The main difference among genomes, transcriptomes, proteomes and metabolomes is their chemical diversity (Figure 1) [13]. Both transcriptomes and genomes provide information on the polymeric molecules composed of only four bases, while proteomes deal with the analysis of proteins that are developed by 20 different amino acids. In contrast, the metabolome is exceptionally chemically diverse and contains 1000 to 200,000 different chemical structures [40,41]. Moreover, metabolites are the downstream products of cell metabolism and provide links with many diverse pathways that happen within a cell [13]. Many metabolites are often produced at the same time and the same metabolite can have roles in multiple pathways [42]. Metabolites produced by the cells often provide phenotypic information of the cells in response to different environmental and genetic changes [43]. Therefore, metabolite analysis is very important and provides an integrative overview of the cellular metabolism and phenotypic characteristics of the cells [13].

The metabolomics community has adopted two different ways of determining metabolome of any biological sample. Metabolite profiling is one of them and it is one of the most powerful approaches that is mainly used for untargeted metabolite analysis. In general, an untargeted metabolite profile usually contains information about both identified and unknown compounds [44]. Recently, targeted analyses of metabolites have become popular and are often combined with untargeted metabolomics data. A comparison between untargeted metabolite profiling and targeted analysis is given in Table 1. Both of these tactics are extensively used for the metabolite analysis of complex samples such cells, blood, urine and beverages.

**Table 1.** Different approaches of metabolite analysis.

| Approach | Advantages | Disadvantages |
|---|---|---|
| Targeted metabolite analysis | • Low limit of detection <br> • Usually quantitative <br> • Data analysis and interpretation are easier <br> • Metabolite data can be connected with pathways | • Limited number of compounds can be targeted <br> • Non-targeted compounds are not considered <br> • Purified standards of targeted compounds are required for quantification |
| Untargeted metabolite profiling | • Unbiased and comprehensive <br> • High-throughput <br> • Allows the discovery of new compounds not expected to be in the sample or not expected to be associated with the biological question | • Semi-quantitative <br> • Larger number of false positives and false negatives <br> • Many unknowns <br> • Data interpretation can be challenging |

This information was collated from [13,15,16,18].

## 2.1. Development of Sensitive and Reproducible Separation and Detection Techniques

During the last 10 years, the field of metabolomics has achieved a very significant improvement in terms of the analytical capability, particularly MS technologies. Now, it is possible to measure as many metabolites as possible using only a minimal amount of samples with high-throughput and exceptional sensitivity [45]. In mass spectrometry, samples can be introduced in different ways and sometimes chromatographic separation (e.g., GC, liquid chromatography and capillary electrophoresis) is a preferred method to allow maximum separation of metabolites in a complex biological sample [13]. However, direct infusion (DI) is also widely used for metabolite profiling, which is usually referred to as metabolic footprinting or fingerprinting depending on whether the analysis is of extra- or intracellular metabolites [43,46]. Due to the development of interfacing systems like atmospheric pressure ionization (API), DI-MS can be used to analyze a sample to obtain mass spectra of metabolites within a few seconds [43]. The requirement for a small amount of sample is the major advantage of using DI-MS. Moreover, no derivatization is required for this analysis and more metabolites are detected by DI-MS compared to GC-MS, making this technique best suited for high throughput non-targeted metabolite profiling [47]. However, DI-MS shows poor reproducibility when analyzing complex mixtures due to the matrix effect. The identification of metabolites by DI-MS is also very troublesome and stereoisomers cannot be resolved using this technique [43,48,49].

A variety of ion sources are available for MS: electrospray ionization (ESI), electron impact ionization (EI), chemical impact ionization (EI), matrix assisted desorption ionization (MALDI), thermospray ionization, atmospheric pressure chemical ionization (APCI), fast atom bombardment (FAB) ionization, field desorption ionization, etc. Among these, EI and ESI are the most commonly used in metabolomics [50]. The mass analyzers that have also advanced significantly and that are widely used by the scientific community are: quadrupole (Q), quadrupole ion-trap (QIT), time of flight (ToF), orbitrap, ion mobility spectrometry (IMS) and fourier transform ion cyclotron resonance (FT-ICR). Quadrupole mass analyzers are very robust, low cost and simple to use, but they offer lower mass resolution and accuracy compared to other mass analyzers [43]. On the other hand, ToF, FTICR and orbitrap, are considered extraordinary instruments that offer the highest mass resolution among all other mass analyzers [13].

NMR is another analytical instrument that has been extensively used by the metabolomics community particularly for untargeted metabolite profiling of complex mixtures (i.e., fruit juices, wines, spirits, urine and blood) [35,51–54]. The efficacy of NMR spectroscopy has been increasingly renowned for its non-invasiveness (non-destructive), throughput and linearity [55]. Moreover, NMR spectroscopy also provides structural, chemical-kinetics and other information in multidimensional applications [56]. Thus, high resolution NMR spectroscopy along with multivariate data analysis has been used for direct characterization of fruit juices, wine [54,57,58], grape berry [59,60] olive oil [61,62] and beer [63,64]. To obtain a global metabolite profile of a complex samples, NMR needs to

be coupled with another non-targeted analytical approach (e.g., MS) [65]. Recently, Bruker developed and launched an instrumentation platform, scimaX MRMS, that combines the capability of NMR and MS and provides superior resolution and mass accuracy albeit the high expense.

*2.2. Advancements in Data Analysis Pipelines*

Mirroring the advancements of the analytical instrumentation platforms, metabolomics data analysis pipelines have also been improved significantly within last 10 years. Particularly, many efforts have been made to make data analysis more efficient and user friendly by a few prominent research groups [45,66,67]. In metabolomics, a few steps are usually involved in the whole data analysis process after raw data are generated using a suitable instrument. As such, raw data need to be preprocessed and annotated prior to statistical analysis. Post-processing steps including data filtering, imputation, normalization, data centering, scaling and transformation are also undertaken [68]. Either in-house or publicly available or commercial software or tools are used for all these steps. Therefore, data analysis generally requires a considerable amount of resources including the time of researchers, purchasing a suite of commercial software or developing the in-house tools.

The type of data analysis software or tools that need to be used generally depends on the instrumental approach used to generate the data. For instance, data generated by NMR usually are processed by specific tools designed for aligning and annotating NMR spectra and are provided mainly by the instrument manufacturers [69]. Databases also are built based on type of samples analyzed to facilitate the process of identifying particular bin/s within NMR spectra [69,70]. Similarly, many software and tools also have been developed for the analysis of GC-MS and LC-MS data [71,72].

Open source software and web interfaces are now providing much better platforms for data analysis, starting from data mining to data interpretation [67,73]. Spicer, Salek, Moreno, Cañueto and Steinbeck [68] recently published a review article stating most of the freely available software tools for metabolomics data analysis. Their review covered tools that are used for data pre-processing, annotating, post-processing and statistical analysis, and readers are advised to consult that review to obtain a wide overview of the open source software [68]. Most of the available software tools are either R based [71,74–76] or Python based [72,77,78].

Metabolomics data analysis, like any other omics approaches, is moving towards cloud-based analysis. According to Warth, et al. [79], cloud computing provides multiple advantages over downloadable desktop based software mainly because of the straightforwardness of data sharing, transferring, managing and archiving. They also reinforced the fact that cloud computing allows a better standardization of data formatting and distribution in addition to ensuring global access of the data without the need for confined high-end computational hardware [79]. However, this process does not come without challenges as it requires a consistent and fast internet connection and often may face security issues in terms of intellectual property [79]. Regardless of the associated risks, cloud computing is becoming very popular for metabolomics data analysis and there are already platforms that make use of this approach, such as XCMS online [67] and Metaboanalyst [80]. Researchers across the different continents are now making use of these cloud-based metabolomics data analysis pipelines. For instance, Metaboanalyst was used by 60,000 researchers from 2000 cities around the world over the past 12 months, and approximately 6000 jobs per weekday or 150,000 jobs/month are usually submitted to this web interface [80]. XCMS online has over 4500 registered users from 120 different countries [67]. These data clearly indicates the popularity and usefulness of cloud-based metabolomics data analysis. The rise of cloud-based data analysis is not only allowing us to handling more data with ease, it is also helping us to tease out the biological meanings from the metabolomics data. However, we are still far from unravelling the true potential of all the datasets available within metabolomics community.

Continuous development of analytical instrumentations and data analysis platforms together is now providing us access to enormous amount of metabolomics data. This brings forward another important issue, and the metabolomics community is now discussing how to manage the

openly available metabolomics data [81]. Community-based initiatives (e.g., MetaboLights and Metabolomics Workbench) develop tools that would allow the storing and exchanging the huge amount of heterogeneous data [82,83]. In addition, efforts of standardizing data sharing and reporting also began in 2007 by a metabolomics community driven initiative, the Metabolomics Standards Initiative (MSI) [84]. Recently, another initiative, COSMOS (Coordination of standards in metabolomics), has started its journey to fill the existing gaps in data reporting and sharing by taking some examples from other omics approaches [85]. Therefore, the metabolomics community is well aware of existing and upcoming challenges due to the omics revolution. This will in turn encourage the further improvements of data analysis pipelines.

## 3. Application of Metabolomics in Grape and Wine Research: State of the Art

The analysis of grape juice or wine samples can be problematic due to the complex matrix arising from either a high sugar or alcohol content. The detection of compounds present at very low concentrations in grape juice and wine can be hampered. Therefore, the matrix effect (ME) may result in poor and unreliable data as it has significant effect on the reproducibility, linearity and accuracy of the methods used by various analytical instruments [86]. ME is a key concern for the analysis of complex biological samples by LC-MS and many studies already have been undertaken to address this issue [87–93]. A sample clean-up step using SPE or SPME or liquid extraction is usually performed to avoid or reduce ME prior to analysis of samples by other methods [93]. More efficient chromatographic separation is also suggested by Trufelli, Palma, Famiglini and Cappiello [86]. However, these pre-analytical steps are time-consuming, arduous and often can cause loss of analytes, which is not appropriate for an unbiased profiling approach [86,87,94]. Details on sample preparation of grapes, wines and related microorganisms are provided in Lloyd, Johnson and Herderich [38].

Despite ME being a major issue for the analysis of grape juices and wines, comprehensive metabolite profiling is becoming an important tool these days. This approach has been efficaciously applied to distinguish white wines [95], to observe vintage effects on juices [96] and also to obtain information about grape chemical composition, wine typicity and quality [97]. Howell, et al. [98] also used metabolite profiling by GC-MS to determine the connections of different *Saccharomyces* species during wine fermentation. Comprehensive metabolome analysis of Sauvignon blanc grape juices and wines also revealed some new insights on the relationship of juice metabolites with key wine volatile metabolites [34]. Hence, untargeted metabolite profiling is indeed a favorable tool in grape and wine research. In addition, many researchers are using a targeted approach to determine specific groups of metabolites in grape juices, wines and wine yeasts [31,99,100]. However, it is noteworthy that even a targeted analysis using a high-resolution analytical platform is able to provide information of over hundreds of metabolites. For instance, lipidomics is one of the branch from targeted metabolomics that allows the determination of wide ranges of lipid species and fatty acids (often over 500) using a suitable analytical instrument [101]. On the other hand, an untargeted approach is gaining popularity for the analysis of volatile compounds in different biological samples, including wines, and often, this area is referred to as a "volatilome" [30,102–104]. Table 2 represents the comparisons between commonly used analytical instruments and their application in grape and wine research [13].

One of the major outcomes from metabolomics is the development of methods for the analysis of different groups of metabolites in complex grape and wine samples [33]. While targeted metabolomics led to the development of methods suitable for the analysis of particular group of metabolites with accuracy and sensitivity, untargeted metabolomics is now enabling us to detect thousands of metabolites just in a single run. Metabolomics has been applied to different areas of agriculture and food sciences albeit the number of publications is much lower compared to other areas (e.g., biomedical, cancer research). Based on the data obtained from Web of Science (on 4th October, 2018), a total of 198 and 154 articles (research and review) have been published in wine and grape metabolomics, respectively, within the last 13 years (Figure 2).

**(a)**

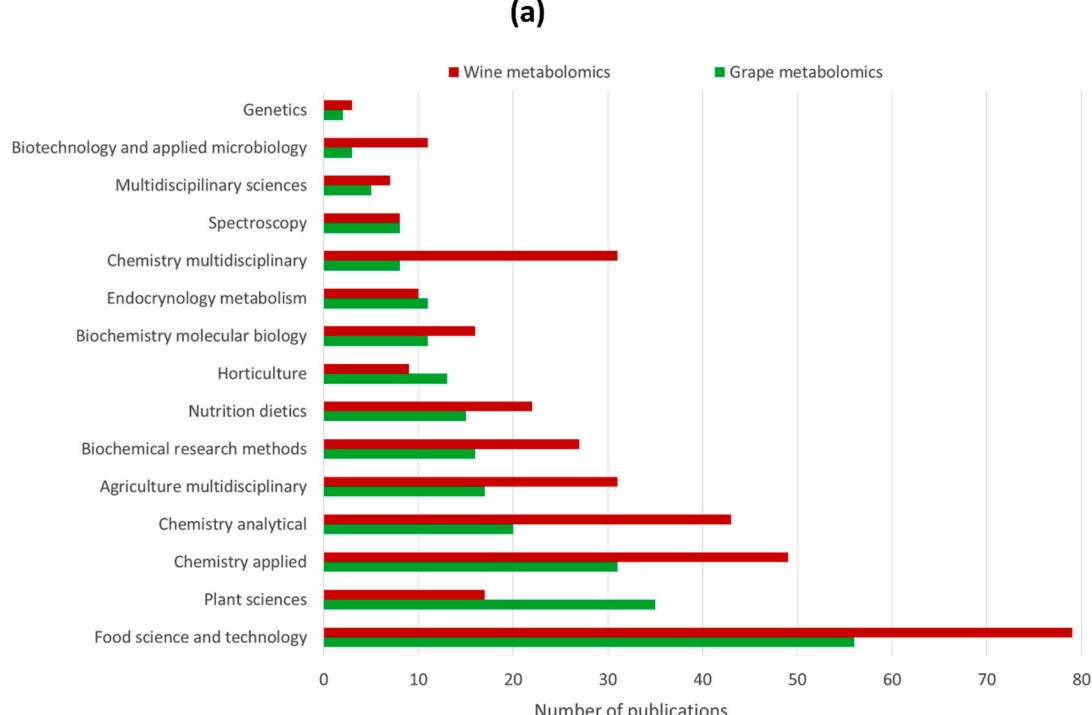

**(b)**

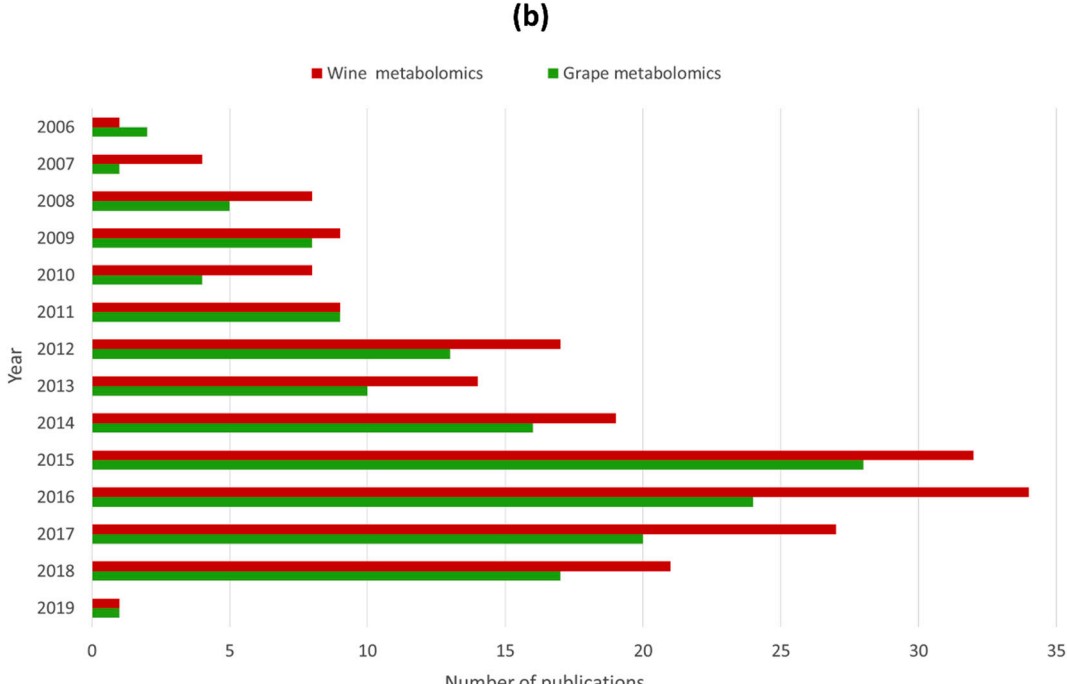

**Figure 2.** Publications on grape and wine metabolomics based on different research areas (**a**) and years (**b**). Data were obtained from the Web of Science.

**Table 2.** Comparisons among different analytical instruments used in targeted and untargeted metabolite analysis of grape juices and wines.

| Analytical Technique | Advantages | Disadvantages | Use in Grape and Wine Research |
|---|---|---|---|
| GC-MS | • High chromatographic resolution<br>• Sensitive and robust<br>• Simultaneous analysis of different groups of metabolites<br>• Large linear range, availability of commercial and in-house MS libraries | • Derivatization is required for non-volatile metabolites<br>• Unable to analyze thermolabile compounds | [8,30,31,34,98–110] |
| LC-MS | • High sensitivity, Derivatization not usually required<br>• Large sample capacity<br>• Thermo-labile compounds can be analyzed | • Average to poor chromatographic resolution<br>• De-salting may be required<br>• Limited commercial libraries<br>• Tough restrictions on LC eluents<br>• Matrix effects | [36,111–119] |
| NMR | • Rapid analysis<br>• Non-destructive<br>• Minimal sample preparation<br>• Quantitative | • Low sensitivity<br>• More than one peak per component<br>• Identification is laborious due to complex matrix | [59,60,95,107,120–123] |
| CE-MS | • High resolution, small volume of sample required<br>• Rapid analysis<br>• Usually no derivatization required | • Poor reproducibility<br>• Poor sensitivity<br>• Buffer incompatibility with MS<br>• Difficulty in interfacing with MS<br>• Limited commercial libraries | [124,125] |

Interestingly, approximately 70% of these articles were published within last five years, which clearly shows that grape and wine researchers are more interested in adopting metabolomics either in a targeted or in an untargeted manner (Figure 2). Based on this published research, some of the most interesting applications are discussed below mainly by highlighting those published in last five years. Readers are also recommended to consult three other review articles where the potential applications of metabolomics in different aspects of grape and wine research have been discussed [35,37,38].

*3.1. Untargeted Metabolomics As A Hypothesis-Generating Tool in Grape and Wine Science*

The journey of metabolomics as an omics tool started over 20 years ago, mainly as a data- and question-driven approach. Therefore, most of the earlier publications in grape and wine metabolomics also aimed to determine as many metabolites as possible in order to develop some new insights into grape growing and winemaking [34,105,117,120]. In 2008, a technology feature was published in Nature that coined a word "wine-omics" to discuss a research project by Kirsten Skogerson at the University of California, Davis, where metabolomics was used as a data-driven approach [126]. Since then, many others adopted untargeted metabolomics to generate hypotheses regarding the role of wine yeasts and grape juice components in the development of wine aroma compounds. For instance, both GC-MS- and NMR-based methods were developed for the untargeted analysis of Sauvignon blanc grape juices and wines [34]. Combined data allowed the authors to generate some data-driven hypotheses for the role of different juice metabolites in the major wine aroma compounds (varietal thiols). The results from a simple juice manipulation experiment confirmed the hypotheses and showed the capability of metabolomics as a hypothesis-generating tool [34]. Similarly, Arapitsas, et al. [127] applied an untargeted LC-MS based metabolomics platform to study the role of micro-oxygenation during winemaking. They formulated hypotheses on the development and reactivity of wine pigment and the role of different primary and secondary metabolites in this matter, thus revealing the benefits of using unbiased, untargeted metabolomics to advance their understanding of wine chemistry [127].

Since the beginning, there was an ongoing need to develop improved analytical platforms that would allow the determination of a large number of metabolites within the grape and wine metabolome. The use of high-resolution MS instrumentations, such as, fourier transform mass spectrometry (FTICR-MS) and ultra-high performance liquid chromatography coupled with quadrupole time-of-flight mass spectrometry (UPLC-Q-ToF-MS) in wine analysis in an untargeted manner is now making it possible to detect metabolites with high precision and mass accuracy [32,116–118,128]. In addition to revealing the true complexity of the wines, this approach is now adding another dimension in terms of obtaining exact mass for formula calculation with retention time information of unknown molecules [117]. Liu, Forcisi, Harir, Deleris-Bou, Krieger-Weber, Lucio, Longin, Degueurce, Gougeon, Schmitt-Kopplin and Alexandre [128] also applied untargeted metabolite profiling using FTICR-MS and UPLC-Q-ToF-MS to determine the outcomes from the interaction of malolactic bacteria and yeasts that either stimulate (MLF+) or inhibit (MLF−) malolactic fermentation. In this study, they were able to detect 3000 discriminant masses that characterized the phenotypes of both MLF+ and MLF− yeast strains in addition to determining MLF− biomarkers. A combination of both targeted and untargeted metabolomics approaches was also found to be beneficial in determining the role of ethanol stress in an off-odor producing yeast, *Dekkera bruxellensis,* and also in generating new knowledge on this contaminant yeast [129].

An untargeted metabolomics approach usually provides an opportunity to look at the system in a holistic way and encourages thinking outside the box and not to be reductive. Therefore, the application of this approach generated a mammoth amount of data in grape and wine research [33,34,109,130]. In the near future and with the development of suitable data analysis platforms, these data can be explored to their full potential. Thus, it shows the promise of new innovation and generation of new knowledge to fill the current gaps.

## 3.2. Study of Terroir, Authenticity and Originality of Grapes and Wines Using a Metabolomics Approach

Wine is a comparatively expensive commodity in modern society, and winemaking is considered not only a science, but also an art. Therefore, both originality and *terroir* are important aspects for the wine producers, particularly for wine makers from old world countries (e.g., European countries). Metabolomics approaches have been applied in this area of wine science to provide analytical tools that would allow differentiation among different wine growing regions [59,118,131], quality control and authentication of wines [55,132]. In a review article published in 2015, Alanon, Perez-Coello and Marina [35] provided a comprehensive discussion on the application of metabolomics by using different instrumentation platforms on wine traceability. Therefore, this section will mostly highlight the works in this subject matter published from 2014 until now.

NMR was the instrument of choice for most of the studies that dealt with *terroir*, authenticity and originality of grapes and wines [58,60,123,131] and Amargianitaki and Spyros [132] provided an excellent overview of application of NMR-based metabolomics in this area. Some of the most recent studies [60,133] that made use of NMR-based metabolomics again re-inforce the fact that NMR is a powerful instrumental approach with high reproducibility and requires minimum sample preparation [55]. For instance, Cassino, Tsolakis, Bonello, Gianotti and Osella [133] applied $^1$H NMR-based metabolomics and chemometrics to differentiate the grapes produced within the Barbera regions of Italy. In addition, they also determined the influence of different climatic factors on the wine composition. One of the interesting studies from Picone, Trimigno, Tessarin, Donnini, Rombola and Capozzi [60] reported the differences among grapes produced by different cultivation systems (biodynamic and organic) using comprehensive NMR analysis. They found a lower amount of sugars, coumaric and caffeic acids and higher concentrations of proline, valine and γ-aminobutyric acid (GABA) in biodynamic grapes than in organic ones. These results clearly indicate that cultivation practices alter the grape metabolome and as a result will also have significant effect on wine quality. Although NMR is widely used for grape and wine analysis, it is noteworthy that this instrument is mainly capable of detecting metabolites that are usually present in higher concentrations [34].

In comparison to NMR, MS provides far better coverage of metabolites present in any biological samples and many studies have been published last five years to demonstrate the application of MS-based metabolomics in determining growing regions of grapes and wines. For example, Roullier-Gall, et al. [134,135] developed analytical platforms using high-resolution MS techniques that were able to detect over thousands of features in grape and respective wine samples. Using these data, they showed the effects of geographical location and vintages on the grape and wine composition produced in Burgundy regions. Bokulich, et al. [136] reported another interesting study demonstrating the relationship between berry microbiome and metabolome and their combined effect on wine *terroir*. They surveyed over 200 commercial wine fermentations within Napa and Sonoma wine counties and determined the wine metabolite profiles using UHPLC-QTOF-MS. Using machine learning models, they showed that the bacterial and fungal consortia in wines correlate with the chemical composition of the finished wines, thus directly influencing the regional characteristics of the wines [136].

A combination of different MS-based metabolomics approaches also seemed to be successful in finding out the terroir effect on grapes and wines. Anesi, et al. [137] applied both GC-MS- and LC-MS-based metabolomics to analyze grape berries from a single clone of the Corvina variety grown in seven different vineyards, located in three macrozones, over a 3-year trial period to determine the effect of *terroir*. Their results showed that the berry metabolome is mainly affected by the vintage. While some of the non-volatile (e.g., stilbene, anthocyanins and flavonoids) and volatile metabolites showed a trend of plasticity over the three vintages, other metabolites including procyanindins and flavan-3-ols seemed to be much more stable.

### 3.3. Study of Yeast Metabolism and Aroma Compound Development during Wine Making

The use of both targeted and untargeted metabolite analysis provides a snapshot of any microbial metabolism based on the growing environment [108,138]. The fermentation process during winemaking is mainly dominated either by a single inoculated commercial wine yeast strain or by a number of wine yeasts already present in the grape juices [139]. Regardless of the type of fermentation, the environmental condition is generally not favorable for any types of yeasts considering grape juice is a high sugar (hyperosmotic stress) and low nitrogen (nutrient limited) growth medium [140] (Figure 3). Moreover, once the fermentation begins and ethanol is produced, wine yeasts go through oxidative stress and ethanol toxicity [108] (Figure 3). As metabolomics allows the analysis of hundreds of metabolites in a single run, it is an excellent tool for the study of the metabolic behavior of wine yeast strains.

Over the past decade, many metabolomics studies have been carried out to determine effect of juice or growth media composition on overall wine yeast metabolism, with particular attention on the developments of fermentation end products [107,108,141]. For instance, two publications from a research group of the University of Auckland reported changes before and after fermentation in Sauvignon blanc juices and how juice composition influences the major varietal aroma compound production by *Saccharomyces cerevisiae* EC1118 [34,107]. Based on a combination of comprehensive metabolite analysis by two different analytical platforms (GC-MS and NMR), they analyzed 63 grape juices and respective wines produced over six different seasons and showed that assimilation of different nitrogen and carbon sources by EC1118 depended on the overall grape juice composition [107]. Moreover, their studies provided some new insights into the metabolism of a wine yeast strain and generated new hypotheses about the potential roles of juice metabolites on the development of varietal aroma compounds. Their data together with other published information provided knowledge on how the biosynthetic pathways of secondary metabolites (e.g., thiols) are expected to be highly interconnected to primary central carbon metabolic pathways. Therefore, any alterations in one or more of these primary metabolic pathways are likely to influence the biosynthetic tariffs of secondary metabolites (Figure 3).

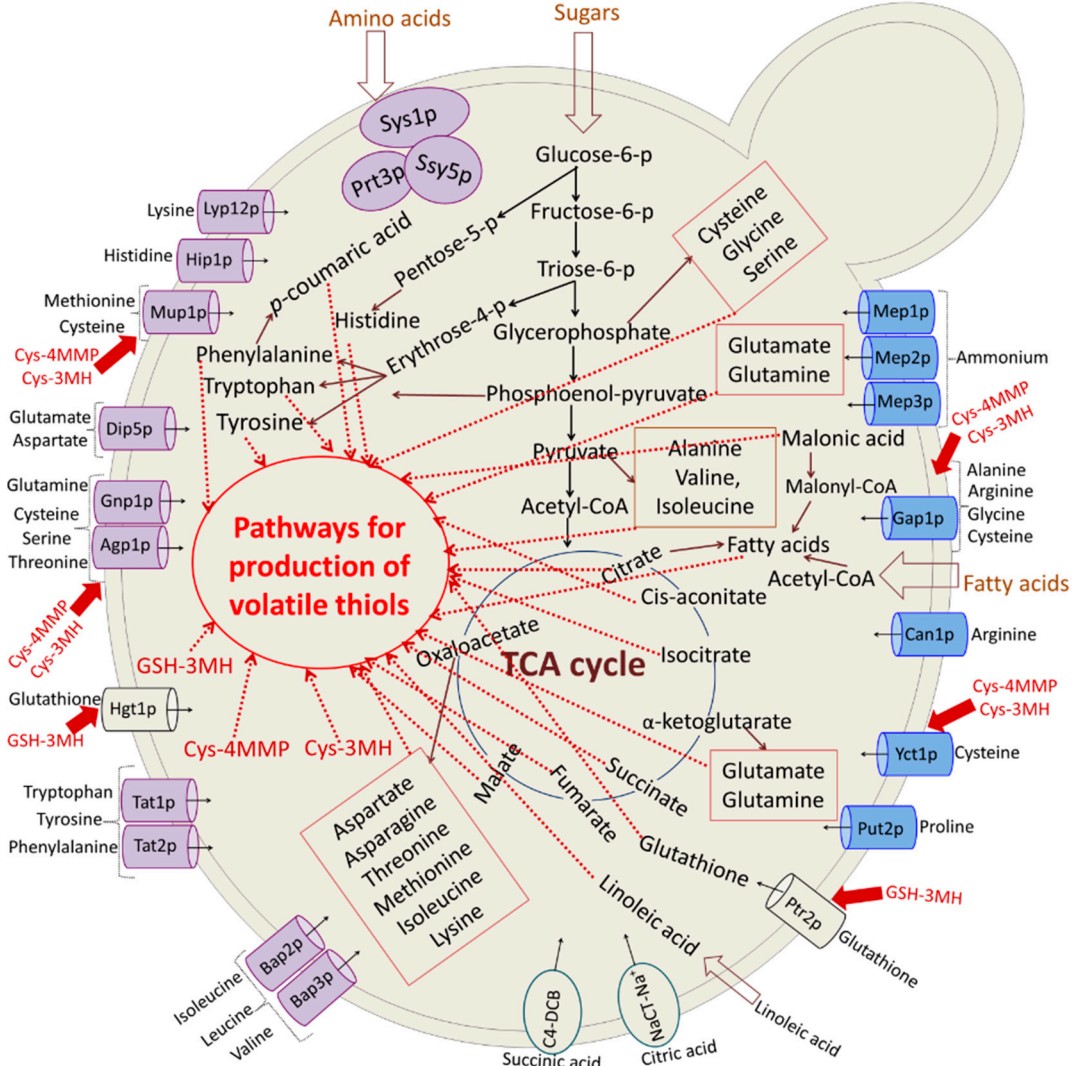

**Figure 3.** The anticipated metabolic network related to varietal thiol pathways in *Saccharomyces cerevisiae* based on the metabolomics data published by Pinu, Edwards, Jouanneau, Kilmartin, Gardner and Villas-Boas [34]. This figure shows the transport mechanisms of different metabolites into the yeast cells. The amino acid transporters regulated by Ssy-Ptr3-Ssy5 (SPS) genes are shown in purple and the transporters regulated by nitrogen catabolite repression (NCR) are shown in blue. It also presents the major pathways of yeast metabolism and how they are connected to the formation of volatile thiols. The red (solid) arrows indicate the suggested ways for the transportation of putative precursors to the yeast cells, while the dashed red arrows indicate which metabolites (e.g., nitrogenous compounds, organic acids and fatty acids) influence the production of varietal thiols in Sauvignon blanc wines. The information for this figure was collated from Pinu [13], Aliverdieva, et al. [142], Crépin, et al. [143], Cooper [144], Hofman-Bang [145], Henscke and Jiranek [146]. Here, cys-3MH and cys-4MMP denote the cysteinylated precursors of 3-mercaptohexanol (3MH) and 4-mercapto-4-methylpentan-2-ol, respectively, and GSH-3MH is the glutathionylated precursor of 3MH.

The influence of different groups of metabolites, particularly amino and fatty acids, and fermentation conditions on winemaking and also on wine yeast metabolism is a well-studied area [108,141,147–151]. A considerable amount of new knowledge has been generated in this matter mostly by using targeted metabolomics or a metabolite-analysis approach. Most of these studies were performed either using a natural grape juice medium [108,141] or grape juice-like synthetic media [149,150]. Casu, Pinu, Fedrizzi, Greenwood and Villas-Boas [108] investigated the effect of pre-fermentative supplementation of an unsaturated fatty acid, linoleic acid, on the metabolism and

aroma compound production by three different wine yeast strains. Using a GC-MS-based metabolomics approach, they showed that increased concentration of linoleic acid affected different primary (e.g., amino acids) and secondary metabolites (e.g., varietal thiols and acetate esters) in a strain-specific manner. Another experiment from the same laboratory also revealed that pre-supplementation of different saturated and unsaturated fatty acids during Sauvignon blanc fermentation significantly affected the metabolism of wine yeast and as a result, end product formation [141]. These metabolomics studies together generated an awareness for the wider wine industry to select wine yeast based on the juice composition in order to obtain wines with desired aromatic profiles.

The effect of temperature and micro oxygenation/oxygen impulse on wine yeast metabolism is another area of wine science where metabolomics was used as a tool [150,152]. For example, López-Malo, Querol and Guillamon [150] compared the metabolome of commercial wine yeast strains (*S. cerevisiae*, *S. bayanus var. uvarum* and *S. kudriavzevii*) at 12 °C and 28 °C in a synthetic grape must medium. Their data confirmed that cryotolerant yeast species (*S. bayanus var. uvarum* and *S. kudriavzevii*) responded differently to the temperature difference than the *S. cerevisiae* strain and the main difference was observed in carbohydrate metabolism. Moreover, an elevated shikimate pathway activity was found in *S. bayanus var. uvarum,* while NAD$^+$ synthesis increased in *S. kudriavzevii* in response to cold temperature. Another study from Rollero, Bloem, Camarasa, Sanchez, Ortiz-Julien, Sablayrolles, Dequin and Mouret [152] reported the development of a model to predict the combined effect of nutrition (nitrogen and lipid) and temperature on the production of fermentative aroma compounds by *S. cerevisiae* EC1118 during winemaking. Their results again proved the complex effect of different environmental parameters on non-volatile and volatile metabolites and shed new light on the synthesis and regulation of secondary metabolites.

### 3.4. Combination of Metabolomics and Transcriptomics to Unravel New Knowledge

As a metabolomics approach provides information on the most downstream products of a cell system, the combination of this tool with other omics approaches can be extremely powerful and may allow the generation of unique knowledge or help to fill the knowledge gaps [139]. Many multi-omics integration studies have already been performed to elucidate more details of *S cerevisiae* metabolism as a cell factory. This integrated approach is still under-utilized in grape and wine research compared to other food and agricultural sectors and only few publications can be found. However, a combination of metabolomics and transcriptomics has been adopted to study different wine yeast strains under wine fermentation-like conditions to develop new insights into the role of oxygen impulse on *S cerevisiae* wine strains [153,154] and to explore the aroma profiles of wines produced by different *S cerevisiae* strains of diverse origins [155].

Stuck fermentation can be a huge problem in winemaking and various technologies are already in use to re-start the fermentation process. Among those, oxygen impulse is often used to avoid the loss of a whole batch of wines. Using integrated metabolomics and transcriptomics, Aceituno, Orellana, Torres, Mendoza, Slater, Melo and Agosin [153] investigated the effects of different ranges of dissolved oxygen on the metabolism of *S cerevisiae* EC1118 grown under a carbon-sufficient but nitrogen-limited medium. They showed that an increase in dissolved oxygen from 1.2 to 2.7 μM caused the yeast cells to change their metabolism from a fermentative to a mixed respiro-fermentative one, which was characterized by a shift in the process of the tricarboxylic acid cycle (TCA) and an activation of NADH transferring from the cytosol to mitochondria. They also observed a significant change in several key respiratory genes, and also in genes related to proline uptake, cell wall remodelling, and oxidative stress. In addition, their results indicated that respiration was accountable for a large portion of the oxygen response in yeast cells during alcoholic fermentation. Another study from the same group also showed the physiological response of *S cerevisiae* EC1118 strain to the sudden increase of dissolved oxygen in a carbon-sufficient but nitrogen-limited medium [154]. Their results confirmed the induction of genes related to mitochondrial respiration, ergosterol biosynthesis, and oxidative stress and the repression of mannoprotein coding genes in response to the increased amount of dissolved oxygen

in the growth medium. However, the authors highlighted the fact that oxygen plays a dual role in winemaking considering some potential detrimental effects on wine aroma, although their integrated omics approach provided some new knowledge on the beneficial influence of oxygen availability on wine yeast metabolism [154].

Integration of transcriptomics and metabolomics data can also be used for the generation of new insights into the yeast metabolism and fermentation process, particularly aroma compound development. For instance, Mendes, Sanchez, Franco-Duarte, Camarasa, Schuller, Dequin and Sousa [155] performed a comparative transcriptomics and metabolomics analysis of four yeast strains from diverse origins (e.g., wine, sake, cacha double dagger and laboratory) at two time points. They used a multivariate factorial analysis to identify new markers that can be used for improvement of aroma production. The combined omics data allowed them to differentiate all the yeast strains at both the metabolic and transcriptomic level and extended their knowledge on the production of wine aroma and flavor. They also identified new genes associated with the development of flavor active compounds, primarily those related to the production of fatty acids, and ethyl and acetate esters [155].

### 3.5. Application of Metabolomics to Study Grape Growth Developments and Grape Vine Disease

Metabolomics is comparatively a new tool for viticultural studies and only a few publications are available that show the suitability of this omics approach, mostly in a targeted manner to study the vine and grape berry development [59,110,156,157]. Recently, Cuadros-Inostroza, Ruíz-Lara, González, Eckardt, Willmitzer and Peña-Cortés [110] published a GC-MS based metabolomics study that aimed at gathering more information on primary metabolites during the grape berry development in different cultivars. They analyzed grape berry samples of two cultivars across six stages (from flowering to maturity) using an untargeted metabolite profiling approach. They identified 115 metabolites in those samples and showed that the changes in metabolite composition were growth stage specific and particularly more distinct during fruit setting and pre-veraison. Moreover, they performed a network analysis and confirmed again that network connectivity of primary metabolites was stage and cultivar dependent. Therefore, they suggested some association between primary metabolites during berry developmental processes between different grapevine cultivars [110]. Another study from Hochberg, Degu, Cramer, Rachmilevitch and Fait [156] also determined the differences in berry metabolism in Shiraz and Cabernet Sauvignon vines under shortfall of irrigation by determining berry skin metabolite profiles using both GC-MS and LC-MS. They clearly observed different types of effect of water deficit on berry metabolism between these two cultivars and showed that water deficit increased the production of stress-related metabolites (e.g., proline, beta-alanine, nicotinate, raffinose and ascorbate) more in Shiraz than the Cabernet Sauvignon grapes. Moreover, water stress affected the polyphenol metabolism uniquely for each cultivar. This metabolomics study revealed a link between the vine hydraulics and water stress-related differences in berry skin metabolism [156].

In addition to grape berry development, early detection of grape vine disease through metabolomics may become an important area of research. A recent study reported some interesting findings on a grape vine disease (grape vine downy mildew) pathosystem using a multi-omics (genomics, transcriptomics and metabolomics) approach [158]. The combination of multiple omics data allowed the authors to characterize the pathosystem of downy mildew causing pathogen, *Plasmopara viticola*, in a molecular and biochemical level. They also determined a potential RNA-based marker that can be used to screen novel resistant grape varieties. Other earlier work by Benheim, et al. [159] also showed the capability of the metabolomics approach in determining potential biomarkers for an early detection method for phylloxera infestation in grapevine. They identified four flavonoid compounds: isorhamnetin glycoside, rutin, kaempherol glycoside and quercetin glycoside in grape vine leaf using LC-MS and validated the specificity of these compounds against phylloxera through field and glasshouse-based trials against nutrient and water stress. A further NMR based metabolomics study of grape vine leaves from phylloxera infested and uninfested vines also revealed the key metabolic changes that occur in the grape vine [160]. For instance, they found that sucrose,

caffeic acid and quercetin were up-regulated in infested grapevines while glucose and leucine were downregulated. Based on both of these studies, they suggested quercetin as a potential marker that could be used for the early detection of phylloxera infestation in grape vines.

## 4. Future Perspectives

In comparison to classical chemical analysis of grape juices and wines, targeted and untargeted metabolomics approaches are more beneficial as these approaches are able to cover more metabolites, including those present in very small concentrations. Both analytical platforms and data analysis pipelines have improved considerably; therefore, our knowledge on different types of metabolites and related pathways has also advanced considerably compared with the last decade. The application of untargeted metabolomics in grape and wine research resulted in the generation of huge amount of data. However, it is still unfortunate that we are currently unable to identify many of the metabolites present within the grape and wine metabolome. Without identification, the interpretation of the metabolomics data is not complete and sometimes meaningless as it is impossible to connect metabolites without identity with pathways [161]. This somehow caused confusions and many are adopting targeted metabolome analysis of specific groups of metabolites within grape and wine research. With targeted analysis, it is now possible to detect and identify over hundreds of metabolites in a single run, particularly using MS-based analytical techniques.

Further developments of analytical instrumentations and data analysis platforms within the next years will allow the generation of even larger data sets. However, the metabolomics community is now well aware of the upcoming issue of omics data revolution and many research groups are now building platforms to make metabolomics data more accessible and manageable [81]. There is also a growing interest in making metabolite identification more robust [161]. This will indeed support the notion of making more sense out of the data obtained from any metabolomics experiment. For instance, machine learning and deep learning approaches are improving; this will in turn be useful to tease out those ever increasing data sets by determining the key features, thus allowing better interpretation of metabolomics and other omics data [162,163].

Efforts have already been made to create databases of metabolites present within a particular system (e.g., food metabolome) [164]. Similarly, grape and wine metabolite databases will also be publicly available, thus providing more information on how grape juice and respective wine composition varies from season to season and region to region [33]. For example, in New Zealand, a comprehensive research program on Sauvignon blanc grape juices and wines has been conducted with the help of the government, and industry provided funding that built a nationwide collaboration among universities and research institutes. Under this program, a database has been created that contains metabolite and other compositional data sets of over 400 grape juices and wines collected over three different seasons and from all the wine-producing regions. This database also harbors information provided by the grape growers and weather stations. Further work on building a predictive tool for the winemakers is ongoing by using these data sets (data not yet published).

It is clear that the application of single omics is still unable to provide a holistic overview of any biological system. Therefore, combinations or the integration of omics methods is becoming increasingly popular although there are still many limitations. The future of grape and wine research will also benefit from the adoption of multi-omics or system biology approaches [140]. Metabolomics combined with genomics, proteomics and transcriptomics could be an extremely powerful tool to study wine yeast metabolism and the overall fermentation process. Connection of metabolites with related genes, proteins and RNAs could lead to the generation of new knowledge on metabolism of wine yeast and other related microorganisms in addition to discovering or improving different microbial strains via metabolic engineering to be used during fermentation.

The field of genome scale modelling is substantially developed [165], and this platform can be used to connect genes, proteins and related metabolites. A genome-scale metabolic model (GEM) or reconstruction of different industrially significant microbes (e.g., *S cerevisiae*) is already

available [166,167]. Over 12 genome-scale metabolic reconstructions are available for *S cerevisiae* [168]. Professor Eduardo Agosin's research group from Pontifical Catholic University of Chile has done a considerable amount of work on creating GEMs for different wine microorganisms including *Oenococcus oeni* and *Pichia pastoris* [169–172]. Therefore, guidelines are already available and this approach can be easily applied to microbial strains used in winemaking. Predictive tools for the winemakers can be developed by integrating metabolic modelling and metabolomics approaches. The wine research community and wine industry will benefit if such tools could be available to aid decision making on what type of juice or yeast strain should be used for the production of wine styles based on consumers' demand.

## 5. Conclusions

Due to the availability of analytical instruments with high resolution and exceptional sensitivity, the analysis of metabolites either in an untargeted or a targeted manner allows us to determine a large number of metabolites. Data analysis pipelines have also developed significantly; therefore, application of metabolomics in grape and wine research is increasing exponentially. Although the ever-increasing number of unknown metabolites is hindering the data interpretation process, this can be overcome in the near future, and community efforts are already underway to deal with this particular issue. As mentioned earlier, the application of metabolomics in grape and wine research is more recent and many are exploring the potential of this approach. However, metabolomics combined with other omics approaches is becoming extremely useful and can be applied in all sectors of grape and wine research. Thus far, we have already generated a significant amount of data and knowledge on grape and wine production systems. In future, this area will most probably lead towards the development of approaches to combine genome-scale metabolic modelling with metabolomics and/or integration of multiomics.

**Funding:** This research received no external funding.

**Conflicts of Interest:** The author declares no conflict of interest.

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
