# Peer review of "Grape and Wine Metabolomics to Develop New Insights Using Untargeted and Targeted Approaches"

_fermentation, doi:10.3390/fermentation4040092_

Round 1

Reviewer 1 Report

The application of metabolomics in grape and wine research resulted in the production of a huge number of data. An accurate interpretation is required to successfully connect metabolites and pathways. 

This manuscript is a good review to understand the complex relationships between several issues in the winemaking process. 

The integration of multi-omics is a very promising area, and this review will be useful for the readers.  

However, I believe the first 360 lines (items 1 and 2) are historical and technical compilation on the omics technologies. The inclusion of this whole block does not give new information about what has already been published in other reviews, whether oenological or not. 

They could be summarized in a pair of paragraphs.

The content of part 3 already provides a more concrete vision and is very appropriated.

Please, correct Transciptomics in figure 1.

After these amends, I recommend publishing this review.

Author Response

I thank Reviewer#2 for his/her feedback on this review article. Based on Reviewer#2’s recommendation, I have shortened the section 1 and 2 in the revised manuscript and kept only those paragraphs that are necessary to provide a summarized overview on the recent developments of metabolomics instrumentation and data analysis pipelines. I have also corrected “Transcriptomics” in the Figure 1.

Reviewer 2 Report

The review "Grape and wine metabolomics to develop new insights using untargeted and targeted approaches" is relevant to the wine fermentation field and is, in general, correctly written. However, I feel that the introductory part of the metabolomics methodology is too long. The first 10 pages (practically half of the review) almost do not mention specific grape or wine studies, just the metabolomics techniques and their history. I suggest shorten the length of this section.

Other minor comments:

-The references in the tables 1 and 2 appeared either grouped or individually, why?

-page 8, 311: a few steps are usually involved in the whole data...

-page 9, 324: ...are now providing much better...

-Table 2, LC-MS: did the authors mean Thermo labile instead of thermo stable?

-page 11, 402: metabolites in complex grape and wine samples...

-Figure 2: the text indicates (pag 12, 409) that both a and b indicates articles number within last 10 years but b graph includes data from 2006 till 2019. Please correct.

-page 13, 418: to consult three other articles

-page 14, 468: Therefore, both originality and terroir are important

-page 17, 565: and acetate esters) in a strain specific manner.

-Page 17 and rest of the text: the abbreviation of Saccharomyces should be S. and not just S

-Page 17, 572:other areas of wine science where metabolomics was used as a tool.

-Page 18, 611: genes related to mitochondrial respiration

Page 20, 702: "the" has a different font size than the rest of the text.

Author Response

I would like to thank Reviewer#1 for all these constructive comments and agree that the introductory section on metabolomics methodology is long. Therefore, in this revised version of the manuscript, section 1 and 2 have been shortened as per Reviewer#1’s recommendation. Moreover, all the changes have been made that were listed in “Other minor comments”.